# Nature Connectedness Reduces Internet Gaming Disorder: The Chain Mediating Role of Intolerance of Uncertainty and Desire Thinking

**DOI:** 10.3390/bs14090844

**Published:** 2024-09-19

**Authors:** Zihui Yuan, Fang Xu, Qingqi Liu

**Affiliations:** 1Key Laboratory of Adolescent Cyberpsychology (CCNU) and Behavior of Ministry of Education, School of Psychology, Central China Normal University, Wuhan 430079, China; yuan245960567@mails.ccnu.edu.cn (Z.Y.); xufang0313@mails.ccnu.edu.cn (F.X.); 2Department of Psychology, Faculty of Arts and Sciences, Beijing Normal University, Zhuhai 519087, China

**Keywords:** nature connectedness, internet gaming disorder, intolerance of uncertainty, desire thinking

## Abstract

While online gaming has become a choice for relaxation and entertainment in today’s digital age, Internet Gaming Disorder (IGD) has also become a widely concerning mental disorder. Nature connectedness has been found to effectively reduce addiction-related risks and alleviate symptoms of addictive behaviors. It is a relatively lacking but very important factor influencing psychological recovery and regulation in the digital society. This study aims to explore the relationship between nature connectedness and IGD, and the mediating roles of intolerance of uncertainty and desire thinking. A total of 571 young people voluntarily participated in the questionnaire survey. The results showed that: (1) nature connectedness was negatively correlated with IGD; (2) intolerance of uncertainty plays a mediating role between nature connectedness and IGD; and (3) intolerance of uncertainty and desire thinking plays a chain mediating role between nature connectedness and IGD. Analysis of the research results indicates that nature connectedness can effectively reduce IGD and reveal its mechanism of action. The findings provide new insights for the study and intervention of IGD in the digital age.

## 1. Introduction

With the continuous advancement of information technology, using online games for entertainment and leisure has become increasingly common. As of June 2023, the size of online game users in China had reached 550 million, accounting for 51% of all internet users, with over half of netizens using online games in their daily lives [1]. In a 2024 report on the US video game industry, the number of video game users in the United States reached 190.6 million, with 61% of Americans playing electronic games for at least 1 h per week [2]. While people are using games for entertainment, the issue of game disorder is becoming increasingly prominent. The American Psychiatric Association included “Internet Gaming Disorder” in the diagnostic criteria for mental disorders in 2013 [3], and the World Health Organization also included “gaming disorder” in the category of “Addictive Mental Disorders” in 2018 [4]. Internet Gaming Disorder (IGD), as an important subtype of internet addiction, refers to individuals who cannot control, excessively, or compulsively play online games, causing physiological, psychological, and social functions to be impaired [5]. IGD can have many adverse effects on an individual’s health and psychology, such as sleep disorders [6], obesity [7], aggression [8], depression, and anxiety [9], and even potentially cause neurological damage leading to a decline in cognitive functions such as vision, hearing, memory, decision-making, etc. [10,11]. Given the harmfulness of gaming disorders, its influencing factors and mechanisms have received research attention and discussion.

The natural environment includes both objective landscapes and people’s subjective experiences. The Biophilia Hypothesis suggests that evolution determines the relationship between humans and nature, and humans have an innate tendency to integrate with nature. The natural environment conforms to human biophilic nature, making people feel more relaxed and happier, with more physical and mental recovery functions [12]. Nature exposure is associated with significant psychological health benefits [13]. Nature connectedness is an important indicator reflecting contact and connection with nature [14], describing the relationship between an individual and the broader natural world [15]. Previous research has found that nature connectedness is negatively correlated with problematic smartphone use [16] and can effectively improve substance use disorders [17]. In addition, previous studies have found that non-adaptive cognition, motivation, psychopathological characteristics, and desire thinking are closely related to the risk of IGD [18,19,20,21], while self-related factors, relationship support, and environmental factors in the family and school are protective factors related to IGD [20,22,23]. Although previous research has considered the impact of environmental factors, it is mainly focused on the social environment [24], and few studies have focused on the impact of the natural environment on IGD.

In conclusion, this study sets out from the perspective of the natural environment, aiming to explore the relationship between nature connectedness and IGD and to reveal its underlying mechanisms.

### 1.1. Nature Connectedness and Internet Gaming Disorder

Nature connectedness refers to a psychological connection between humans and the natural environment [14]. In modern times, due to the continuous advancement of urbanization and technological development, people have fewer and fewer opportunities to come into contact with the natural environment, gradually distancing themselves from nature and being surrounded in the long term by industrial products. Researchers have reviewed various types of artworks (such as music, novels, films, etc.) from the entire 20th century and beyond, where cultural expressions are gradually moving away from nature, which to some extent reflects that people are gradually disconnecting from nature [25]. The reduction in contact with nature also means a weakening of nature connectedness [26]. As life becomes more convenient, we also lose the benefits brought about by connecting with nature. Nature connectedness is considered as the bond and emotional link between individuals and the natural environment. Research indicates that individuals’ cognitive and emotional experiences of nature can significantly influence their level of intimacy with the natural environment [27]. Through identification with nature and emotional connections, individuals are more likely to actively engage in activities in the natural environment and experience a deep connection with the natural world. This deep connection may lead individuals to place greater importance on the significance of the natural environment, thereby promoting more frequent interactions with nature. Therefore, nature connectedness can promote individuals’ contact with nature, attract the attention of gaming disorder individuals through the automatic and bottom-up features of nature itself, and divert their addictive attention bias, thereby promoting the recovery of psychological resources and alleviating IGD.

Previous research has found that nature connectedness has a positive impact on individual psychological health, such as enhancing life satisfaction [28], improving individual happiness [29,30], and effectively relieving stress and anxiety levels [31]. Researchers have also begun to notice the role of nature connectedness in addiction and mental illness. For instance, research has shown that walking in green spaces can reduce the levels of depression in patients with depression [32]; natural psychological treatment methods have gradually been applied to clinical practice [33,34]. Nature connectedness has also been found to be negatively correlated with smartphone addiction [35] and has a mitigating effect on substance use disorders [17]. However, fewer researchers have paid attention to the relationship between nature connectedness and IGD.

IGD implies being attracted to game-related factors and rewards [36] and directing one’s attention to activities that promote addictive use [37]. This results in the depletion of attention resources and the emergence of maladaptive phenomena. According to the Attention Restoration Theory (ART), there are four components in the natural environment: being away, fascination, extent, and compatibility. Experiencing these and feeling a connection with nature can help restore individual psychological resources [38], specifically, ART suggests that directed attention is a limited resource that may become depleted after long-term and/or intensive use. The characteristics of nature itself can allow individuals to easily shift their attention to the natural environment and spend less directed attention on resources, thereby promoting the individual’s recovery from the state of attention depletion [39].

Building upon the aforementioned theoretical and empirical research foundation, the present study proposes hypothesis 1: there is a negative correlation between nature connectedness and IGD.

### 1.2. The Mediating Role of Intolerance to Uncertainty

Intolerance of uncertainty is a key component of anxiety-related psychopathology, encompassing negative cognitive tendencies relating to uncertain and unknown events, including cognitive, emotional, and behavioral responses [40]. Previous research has found that nature connectedness can help individuals regulate negative emotions [41], but no research has explored the relationship between nature connectedness and this negative cognitive tendency. Nevertheless, it is possible to hypothesize about the potential relationship between nature connectedness and intolerance of uncertainty from both emotional and cognitive perspectives. From the emotional perspective, individual tolerance thresholds for uncertainty vary in the same context, and those who cannot tolerate uncertainty will experience anxiety and depression caused by ambiguity [42]. According to the Biophilia hypothesis [43], humans have a profound connection with nature, and through natural environments, individuals can experience a soothing, restorative, and even healing feeling [44]. Research has also shown that exposure to nature-related elements can reduce anxiety levels [45,46]. Therefore, when individuals are exposed to the natural environment and establish a connection with nature, this restorative experience can impact the current anxiety state of individuals who cannot tolerate uncertainty [47]. From the cognitive perspective, previous studies have found that nature connectedness can positively influence cognitive regulation [48] and is positively correlated with cognitive reappraisal [49]. This means that nature connectedness can help individuals who cannot tolerate uncertainty to change their perception and understanding of uncertain events, adjusting their negative cognitive tendency. Therefore, we believe that nature connectedness can negatively predict intolerance of uncertainty.

Intolerance of uncertainty is also an important predictor of IGD. According to the Interaction of Person–Affect–Cognition–Execution (I-PACE) model [50], the development of addictive behavior is the result of the interaction between susceptibility variables, emotional and cognitive reactions to specific stimuli, and executive function. As part of the emotional reaction in the I-PACE model [51], individuals with a low tolerance for uncertainty may over-worry about threats when facing uncertain events, while also underestimating their ability to handle the threats. This may lead them to experience negative emotions more easily and tend to adopt passive coping strategies, such as using online games as a means of escape to deal with the negative experiences brought about by uncertainty, resulting in the consequences of IGD. Although no research has directly revealed the relationship between intolerance of uncertainty and IGD, there have been numerous studies exploring the relationship between smartphone addiction, internet addiction, and social media addiction. A questionnaire study of 1269 college students found that intolerance of uncertainty had a significant positive prediction effect on smartphone addiction [52]; a survey of 1006 Chinese adolescents found that intolerance of uncertainty was significantly positively correlated with internet addiction [53]; and a survey conducted during the COVID-19 pandemic found that intolerance of uncertainty could significantly positively predict problematic use of social media [54].

Based on the above, this study proposes hypothesis 2: nature connectedness negatively predicts IGD via the mediating role of intolerance of uncertainty.

### 1.3. The Mediating Role of Desire Thinking

Desire thinking, as one of the key factors affecting addiction [55], refers to a conscious and voluntary cognitive process that links triggering factors such as negative emotions, impulsiveness, and cravings to addictive behavior [56]. It is composed of two components: cognitive imagery and verbal perseveration [57]. Cognitive imagery refers to the allocation of attentional resources to goal-related information and expected behavior, and multi-sensory processing through expected positive imagery or positive memory recall related to the goal. Verbal perseveration refers to the prolonged self-dialogue concerning the valuable reasons for engaging in the goal-related activity and the achievements gained from it. Desire thinking has been found to be closely related to various forms of Internet addiction, including IGD [18,58], problematic use of social media [59], and problematic Internet use [60]. According to the ego-depletion model, resisting the desire for the expected goal yields a psychological cost and consumes limited psychological resources, such that the effort to resist temporarily exhausts those resources [61]. Moreover, the process of self-regulatory depletion may be influenced by the maladaptive choice of self-regulatory strategies, of which desire thinking is one [55]. Desire thinking amplifies the craving emotion and allocates attentional resources to goal-related information about addictive activities, leading individuals to believe that achieving the goal is the only way to alleviate their agony [57]. This intensifies problematic use and further leads to gaming disorder.

According to the previous instructions, nature connectedness has advantages in psychological recovery and promoting positive emotions [62]. Therefore, nature connectedness may directly help individuals effectively recover from states of psychological resource depletion, thus directly reducing the generation of desire thinking. On the other hand, previous research has found that natural environments can effectively inhibit individuals’ impulsivity and provide self-control. A survey of 169 boys and girls living in the city center with the same floor height compared the impact of the presence or absence of natural environments near buildings on self-control. The results showed that green spaces outside the home can effectively improve self-discipline [63]. An experimental study compared impulsive decision-making differences among participants in natural, built, and control conditions. The results showed that exposure to a natural environment significantly reduced impulsive decision-making compared to a man-made environment [64]. Impulsivity and self-control are accompanying factors of desire thinking [65]. Therefore, nature connectedness can promote individuals to engage in natural contact behaviors, alleviate impulsivity, and improve self-control, thus controlling and weakening the influence of desire thinking.

Based on the previous evidence, this study proposes hypothesis 3: nature connectedness negatively predicts IGD via the mediating effect of desire thinking.

### 1.4. The Chain Mediation from Intolerance of Uncertainty to Desire Thinking

According to the I-PACE model [50], intolerance of uncertainty can be seen as an internal trigger, as individuals may experience negative emotions and stress reactions when faced with ambiguous situations [51]. These internal triggers can stimulate desire thinking, leading individuals to crave or impulsively seek addiction-related stimuli. When individuals come into contact with nature and establish a connection with it, this restorative experience can influence the current anxiety levels of individuals with low uncertainty tolerance [47], alleviating this anxiety [45], thereby reducing intolerance of uncertainty. The expression of desire thinking in individuals can be viewed as a conditioned response to addiction-related stimuli [51]. Intolerance of uncertainty may lead to negative emotions and anxiety, intensifying the individual’s need for desire thinking, thereby escalating cravings for addiction-related stimuli. This process may reinforce the individual’s pursuit of addictive behavior, ultimately leading to the occurrence and relapse of addictive behaviors.

In addition, the cognitive–behavioral model of pathological internet use proposes that addictive factors can be mainly categorized into proximal and distal factors. The distal factors are those that are to some extent distant from the symptoms and are necessary conditions for addiction. Psychopathology and stress are distal factors that lead to addiction. Intolerance of uncertainty, which is a key component of related psychopathology such as worry and anxiety disorders, makes individuals who tend to experience stress from the uncertainty in vague daily life more likely to have addiction [66]. Therefore, intolerance of uncertainty can be regarded as a distal factor in the model. Proximal factors are those that directly lead to the symptoms and are sufficient conditions for addiction [67]. The most central factor among them is maladaptive cognition, which refers to distorted cognitive and thinking processes. Desire thinking is considered to be a maladaptive, pathologic, and erroneous cognitive process associated with behavioral dysregulation [55]. Therefore, desire thinking can be regarded as maladaptive cognition and a proximal factor in the model. Due to the tendency of intolerance of uncertainty as a distal factor, individuals are more likely to choose online games that have a clear direction and rules and allow complete control of roles and development, which can lead to the creation of desire thinking as a proximal factor. Finally, the combination of distal and proximal factors results in IGD. Nature connectedness can alleviate the negative effects of uncertainty through the restorative effects brought by natural contact [63].

This study proposes hypothesis 4: nature connectedness negatively predicts IGD through the serial mediation chain from intolerance of uncertainty to desire thinking. The model diagram of this study is shown in Figure 1.

## 2. Materials and Methods

### 2.1. Participants and Procedure

In this study, we recruited 580 young people who frequently play online games to participate in a questionnaire survey. The questionnaire was distributed online through internet platforms frequently used by game players. We then screened the questionnaire by excluding participants who selected “I do not play online games” and finally obtained 571 valid questionnaires. Among them, there were 323 males and 248 females, with an average age of 23.42 years (SD = 4.64). Participants spent an average of 22.22 h per week playing games (SD = 18.03). All participants gave informed consent, confirming their voluntary participation and understanding of the objectives and procedures of the study. Participants received a reward of CNY 3 after completing the entire survey. This study has been approved by the ethics committee and followed the principles of the Helsinki Declaration.

### 2.2. Measures

#### 2.2.1. Inclusion of Nature in the Self Scale (INS)

This study used the Inclusion of Nature in the Self Scale developed by Schultz [68]. This scale was adapted from a measure of interpersonal closeness developed by Aron, Aron, Tudor, and Nelson [69] and consists of only one item. The item features seven sets of circles, with one circle representing the self and another representing nature. The circles range from having no overlap to almost complete overlap, indicating the degree of fusion between self and nature. The more overlap between the circles, the greater the degree of fusion between self and nature. Participants can select the appropriate set of circles to represent their relationship with nature. The scale consists of only one item, making it simple and easy to administer. Research has shown that this scale has good validity and a wide range of applications [70,71]. However, due to the scale’s single-item nature, reliability and validity cannot be calculated [69], and completing the test requires a good understanding of the abstract representation of the self–nature relationship, which undoubtedly increases the difficulty for individuals to accurately report this abstract concept of nature connectedness.

#### 2.2.2. Desire Thinking Questionnaire (DTQ)

This study used the Desire Thinking Questionnaire developed by Caselli and Spada [57]. The questionnaire was adapted to apply to gaming scenarios and translated into Chinese following the approach of Grajewski and Dragan [5]. The questionnaire consists of 10 items, such as “If I haven’t played a game for a long time, I think about it constantly”, scored on a 5-point scale (1 = not at all, 5 = completely), with higher scores indicating higher levels of desire thinking. The alpha coefficient of the questionnaire was 0.92. In the revised questionnaire, the Kaiser–Meyer–Olkin (KMO) value was 0.93 and the Bartlett sphericity test resulted in *p* < 0.001, indicating good structural validity.

#### 2.2.3. Intolerance of Uncertainty Scale (IUS)

This study used the revised Chinese version of the Intolerance of Uncertainty Scale, based on Carleton et al.‘s original scale [72]. The scale consists of 12 items, such as “when it came time to take action, uncertainty would hold me back”, rated on a 5-point Likert scale ranging from “completely inconsistent” to “completely consistent”, with higher scores indicating lower tolerance for uncertainty. The scale includes two sub-dimensions: sub-dimension one is inhibitory anxiety, which reflects the tendency for individuals to avoid uncertainty and feel inhibited in their actions; sub-dimension two is prospective anxiety, which reflects anxiety and emotional anticipation for uncertain future events. The alpha coefficient of the scale was 0.87. In the revised questionnaire, the Kaiser–Meyer–Olkin (KMO) value was 0.92 and the Bartlett sphericity test resulted in *p* < 0.001, indicating good structural validity.

#### 2.2.4. Internet Gaming Disorder Scale-20

IGD-20 is a diagnostic tool developed by Pontes and others based on the diagnostic criteria of the DSM-5, which includes six factors and 20 items [73], such as “I often lose sleep because I play online games for a long time”. The Likert-5 scale (1 = strongly disagree, 2 = disagree, 3 = neither, 4 = agree, 5 = strongly agree) is used for rating, and a total score above 71 indicates the presence of IGD. The revised Chinese version of the scale by Qin and others was used in this study [74], with an α coefficient of 0.92. In the revised questionnaire, the Kaiser–Meyer–Olkin (KMO) value was 0.94 and the Bartlett sphericity test resulted in *p* < 0.001, indicating good structural validity.

### 2.3. Data Analysis

In this study, SPSS 27.0 was used as the primary tool for data processing. The preliminary data analysis included examining the common method bias and validating the reliability of the scales. Descriptive and correlational analyses were then conducted to explore the relationships between variables. Building on these initial data explorations, SPSS PROCESS was used to further investigate the linkages among nature connectedness, IGD, intolerance of uncertainty, and desire thinking.

## 3. Results

### 3.1. Preliminary Analyses

In order to address potential common method bias, Harman’s one-way ANOVA was conducted using SPSS to test the self-report data collected in this study. The test found that six factors had eigenvalues greater than 1, with the first factor accounting for 34.02% of the variance. Importantly, this value was below the critical threshold of 40%, indicating that there was no considerable common method bias [75]. Meanwhile, we calculated the Variance Inflation Factor (VIF) in the regression analyses and found that the VIF for the variables were all below 3 (ranging from 1.00 to 2.87), indicating the absence of multicollinearity among the variables [76].

The results of the correlational analysis showed a significant negative correlation between nature connectedness and intolerance of uncertainty, desire thinking, and IGD. In addition, there was a significant positive correlation between IGD and intolerance of uncertainty, and desire thinking. Intolerance of uncertainty was positively correlated with desire thinking. Table 1 provides the mean, standard deviation, and correlation matrix for each variable.

### 3.2. Testing for the Proposed Mediating Model

To examine the mediating role of intolerance of uncertainty and desire thinking, we used model 6 of the PROCESS macro with 5000 bootstrap samples. Furthermore, when testing for social mediation effects, we controlled for gender, age, and gaming duration (average weekly gaming time). Table 2 shows the results of our analysis. The findings of this study suggest that nature connectedness negatively predicts IGD directly (*β* = −0.09, *p* < 0.01) and has a significant negative effect on intolerance of uncertainty (*β* = −0.31, *p* < 0.001). Additionally, intolerance of uncertainty has a significant positive effect on IGD (*β* = 0.21, *p* < 0.001) and a significant positive effect on desire thinking (*β* = 0.40, *p* < 0.001). Desire thinking significantly predicts IGD positively (*β* = 0.63, *p* < 0.001), whereas the effect of nature connectedness on desire thinking is not significant (*β* = 0.01, *p* = 0.76).

The results showed that the total effect of nature connectedness on IGD was significant (effect = −0.22, 95% CI = [−0.31, −0.14]). Furthermore, nature connectedness had a significant indirect effect on IGD through intolerance of uncertainty (effect = −0.06, 95% CI = [−0.10, −0.04]), accounting for 29.37% of the total effect; however, the indirect effect of nature connectedness on IGD through desire thinking was not significant (effect = 0.01, 95% CI = [−0.04,0.05]). The serial mediation effect was significant (effect = −0.08, 95% CI = [−0.11, −0.05), accounting for 35.36% of the total effect, which essentially supports our model hypothesis (see Table 3 for specifics). Figure 2 illustrates the effect of nature connectedness on IGD.

## 4. Discussion

The purpose of this study was to explore the relationship between nature connectedness and IGD, and to investigate the mediating role of intolerance of uncertainty and desire thinking in this relationship. Our results supported hypotheses 1, 2, and 4. Nature connectedness can negatively predict IGD directly, and indirectly through the independent mediating role of intolerance of uncertainty and through the chain mediating role of intolerance of uncertainty and desire thinking. These findings have important implications for promoting further research on nature connectedness and intervention for IGD.

### 4.1. The Direct Effect

Consistent with our hypothesis 1, the study results found a significant negative correlation between nature connectedness and IGD. Although there have been no studies directly exploring the relationship between nature connectedness and IGD, many studies have revealed the relationship between nature connectedness and other forms of addiction. For example, Zhu found that nature connectedness is an important predictor of less problematic smartphone use, and manipulating individuals’ connection with nature in experiments showed that participants reported a lower desire to use smartphones in natural (vs. urban) conditions [77]. Wang et al. found a negative correlation between excessive mobile phone use and exposure to nature [21]. Intervention studies have shown that viewing nature as an internal spiritual connection and using it to reorganize cognition to overcome distorted cognition that leads to addiction can effectively alleviate individuals’ substance use disorder [17]. Our study results demonstrate that the recovery benefits brought about by connecting with nature may reduce individuals’ risk of IGD. On the other hand, stress is one of the risk factors for IGD [78]. According to the escapism perspective of IGD, individuals may choose games to escape the stress of real life, attempting to seek temporary comfort in the virtual world and leading to addiction [20]. The stress recovery theory (SRT) in environmental psychology points out that humans’ physiology and psychology have a tendency to adapt to natural environments, which may stem from humans’ innate attention to and active response to natural-related content [79]. As the brain and sensory systems evolved in natural environments, processing natural content may be easier and more effective [38]. In contrast, when faced with non-natural situations lacking nature connectedness (such as urban buildings and internet gaming), individuals may require more resources to cope or adapt. Therefore, higher nature connectedness is more likely to lead to exposure to nature [26], which is beneficial for individuals’ psychological recovery. The psychological recovery characteristics of nature can help individuals alleviate their perception of stress and reduce the risk of IGD caused by escaping stress.

### 4.2. Mediating Effect of Intolerance of Uncertainty

Consistent with the proposed hypothesis 2, the results found that nature connectedness had a negative impact on IGD by reducing the effect of intolerance of uncertainty. Individuals with intolerance of uncertainty tend to feel stressed and uneasy about ambiguous situations in life, trying to avoid uncertain events and considering uncertainty as negative [40]. The ambiguity that often exists in daily life can trigger negative reactions in individuals with intolerance of uncertainty [80], which is usually accompanied by anxiety and stress. According to the stress recovery theory, the level of environmental arousal has an impact on recovery from overstimulation or stress, and in environments with lower levels of arousal, recovery may be faster [79]. In addition, research has also shown that when individuals feel stressed or anxious, their preference for complexity levels decreases, which is consistent with the arousal theory [81]. Considering that natural environments may have lower complexity levels and other arousal characteristics compared to urban environments [81], when individuals face anxiety, stress, and discomfort caused by intolerance of uncertainty, natural environments can effectively mitigate the negative effects. Nature connectedness is an important indicator that reflects the contact and connection between humans and nature [14]. Therefore, individuals with higher levels of nature connectedness means that they have better recovery in the face of stress and anxiety and can alleviate intolerance of uncertainty more effectively.

Intolerance of uncertainty is also a risk factor for IGD, and previous studies have found that intolerance of uncertainty can positively predict many internet-related addictive behaviors [52,53]. According to the compensatory internet use model, the root cause of problematic internet use or addiction is the individual’s negative reaction to life circumstances, and it is compensated for by various forms of internet use [82]. People who cannot tolerate uncertainty lack a sense of control in their lives [83], and online games, where players control their characters and have more agency, may provide a feeling of control, leading them to choose online gaming as a way to alleviate their uncertainty in reality, which may lead to the development of IGD. The lower sense of uncertainty in natural environments may help individuals break free from their dependence on online gaming, thereby relieving IGD.

### 4.3. The Chain Mediating Effect

The research results indicated that intolerance of uncertainty fully mediated the relationship between nature connectedness and desire thinking, which is consistent with our hypothesis 4. According to the I-PACE model, perception of internal triggers (such as negative or positive emotions, stress) or external triggers (such as pictures, keyboard sounds) can elicit response experiences related to addiction-related cues and create cravings for addiction-related cues [84]. Individuals who cannot tolerate uncertainty tend to have negative reactions to ambiguous situations [40], which can result in feelings of anxiety and stress when faced with uncertain events in life. These internal triggers that cause stress and anxiety can further induce the individual’s desire thinking and arouse cravings [85]. Meanwhile, desire thinking can help individuals temporarily regulate the emotional turmoil caused by the disparity between the actual state and the ideal state of the goal (addictive behavior), shifting the attention to the expectation and imagination of the ideal goal [57]. However, the I-PACE model also points out that the experience of satisfaction and compensation will strengthen the experience of craving in the later stages of addiction behavior [86]. This means that the individual’s desire thinking will trigger stronger cravings and lead to the occurrence and relapse of addiction behavior. Some indirect studies in the past have also shown the relationship between intolerance of uncertainty and desire thinking. A survey of opioid addicts found that intolerance of uncertainty was positively correlated with impulsivity and anxiety [87], while a questionnaire study conducted in the community found that intolerance of uncertainty was positively correlated with impulsive behavior associated with externalizing psychopathology and related risks [88]. Impulsivity is a factor that accompanies desire thinking [65]. The positive correlation between desire thinking and IGD has also been directly demonstrated [18]. According to the stress recovery theory, natural environments can help alleviate the stress and anxiety experienced by individuals [79], which can effectively help individuals who cannot tolerate uncertainty regulate negative emotions and alleviate the negative effects of intolerance of uncertainty.

### 4.4. Implications

The theoretical significance of this study lies in providing new research perspectives and ideas on how to alleviate IGD in the context of gradually distancing ourselves from nature in the digital age. It expands the research on the impact of nature connectedness on addiction and provides a deeper understanding of the internal mechanisms of nature connectedness on IGD.

In practice, this study offers meaningful insights to reduce the risk of IGD and alleviate related symptoms. Nature connectedness can directly reduce IGD and also mitigate intolerance of uncertainty and desire thinking. Therefore, it is necessary to enhance individuals’ connection with nature through daily activities or arrangements. Contact with nature can enhance nature connectedness, such as arranging green plants in frequently used areas at home, engaging in activities at outdoor or nature-rich places such as parks, and setting desktop backgrounds as natural environment images. These methods can alleviate the negative effects of intolerance of uncertainty, and can also curb desire thinking, thus alleviating the role of IGD. In clinical treatment, more consideration can be given to nature-based and side-effect-free psychological therapies, such as horticultural therapy and wilderness therapy.

### 4.5. Limitations and Future Research Directions

There are some limitations to this study.

Firstly, the cross-sectional design cannot establish causal relationships. Future studies could manipulate nature connectedness through experimental designs to explore the causal relationships among these variables and conduct intervention studies.

Secondly, non-probability sampling introduces the risk of selection bias, where certain segments of the population are more likely to be included in the sample than others. In the context of researching the impact of nature connectedness on reducing IGD, non-probability sampling may lead to a sample that is not fully representative of the target population. Therefore, while the study provides valuable insights into the role of nature connectedness in mitigating IGD, the use of non-probability sampling methods is a limitation that should be acknowledged. Future research could benefit from employing probability sampling techniques to enhance the representativeness and generalizability of the findings.

Thirdly, using only one item to measure the variable of “nature connectedness” has limitations in terms of reliability and validity. A single item may not capture the complexity and multidimensionality of the concept adequately, potentially leading to measurement error and incomplete assessment of individuals’ true connection to nature. To improve this, future research could consider using scales such as the Connectedness to Nature Scale developed by Mayer and Frantz [14]. By capturing various aspects of individuals’ relationship with nature, this scale can provide a more comprehensive and accurate assessment, leading to more reliable and insightful findings regarding the impact of nature connectedness on various outcomes.

Fourthly, this study only considers the general addiction to online games without distinguishing different types of online games. Different types of online games provide different types of satisfaction to individuals. For example, first-person shooter games or MMOs mainly satisfy players’ sense of achievement through winning, while otome games mainly provide a quasi-social and romantic emotional satisfaction for players. Further research is needed to investigate whether the psychological restoration and emotional regulation brought by nature connectedness can be effective in dealing with different types of gaming disorders.

In addition, it is necessary to explore other potential mediator and moderator mechanisms between nature connectedness and IGD, especially those risk factors that lead to IGD and protective factors that reduce addiction risk.

## 5. Conclusions

Our research found a significant negative correlation between nature connectedness and IGD, emphasizing the important role of nature in alleviating addiction. Additionally, this study explored the mediating effects of intolerance of uncertainty and desire thinking, verifying the mediating effect of intolerance of uncertainty and the chain mediation of intolerance of uncertainty and desire thinking. These results enhance our comprehension of the internal mechanisms underlying the relationship between nature connectedness and IGD by elucidating how they are associated. The findings provide valuable insights for mitigating the risk of IGD and reducing related symptoms.

## Figures and Tables

**Figure 1 behavsci-14-00844-f001:**
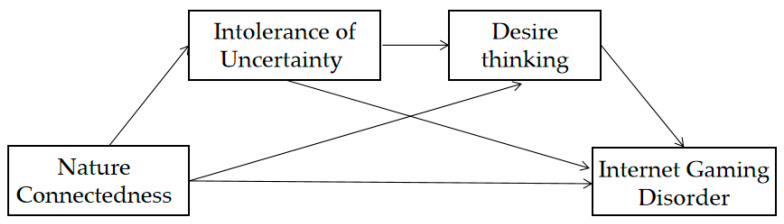
Model hypothetical path diagram.

**Figure 2 behavsci-14-00844-f002:**
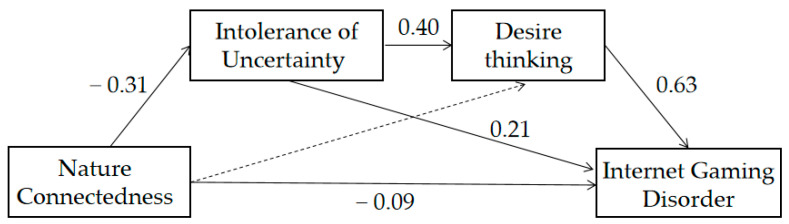
Model path coefficient diagram.

**Table 1 behavsci-14-00844-t001:** Results of correlation analysis between variables.

Variables	M (SD)	1	2	3	4
1. Nature connectedness	4.42 (1.78)	-			
2. Intolerance of uncertainty	38.18 (9.77)	−0.34 **	-		
3. Desire thinking	29.35 (10.11)	−0.10 *	0.37 **	-	
4. Internet gaming disorder	54.54 (14.70)	−0.21 **	0.46 **	0.72 **	-

*n* = 571. * *p* < 0.05. ** *p* < 0.01.

**Table 2 behavsci-14-00844-t002:** Regression analysis.

Model		Fit Index		Significance		
Outcome Variable	Predictor Variable	*R^2^*	*F*	*β*	*t*	95%CI
Intolerance of uncertainty		0.14	22.65 ***			
	Gender			0.25	3.05 **	[0.09, 0.41]
	Age			−0.08	−1.89	[−0.16, 0.00]
	Weekly playtime			−0.02	−0.23	[−0.19, 0.15]
	Nature connectedness			−0.31	−7.61 ***	[−0.39, −0.23]
Desire thinking		0.15	20.17 ***			
	Gender			−0.19	−2.45 *	[−0.35, −0.04]
	Age			0.05	1.23	[−0.03, 0.14]
	Weekly playtime			0.03	0.97	[−0.03, 0.08]
	Nature connectedness			0.01	0.31	[−0.07, 0.09]
	Intolerance of uncertainty			0.40	9.69 ***	[0.32, 0.48]
Internet gaming disorder		0.57	128.58 ***			
	Gender			−0.05	−0.91	[−0.17, 0.06]
	Age			0.04	1.40	[−0.02, 0.09]
	Weekly playtime			0.05	0.37	[−0.20, 0.30]
	Nature connectedness			−0.09	−3.05 **	[−0.14, −0.03]
	Intolerance of uncertainty			0.21	5.75 ***	[0.14, 0.28]
	Desire thinking			0.63	19.08 ***	[0.57, 0.70]

* *p* < 0.05. ** *p* < 0.01. *** *p* < 0.001.

**Table 3 behavsci-14-00844-t003:** Total, direct, and indirect effects.

Path	*SE*	LLCI/ULCI (95%)		Relative Value
		LLCI	ULCI	
NC → IU → IGD	−0.06	−0.10	−0.04	29.37%
NC → DT → IGD	0.01	−0.04	0.05	-
NC→ IU → DT → IGD	−0.08	−0.11	−0.05	35.36%
Total Effect	−0.22	−0.31	−0.14	-

Note. NC, nature connectedness; IU, intolerance of uncertainty; DT, desire thinking; LLCI, lower limit confidence interval; ULCI, upper limit interval.

## Data Availability

The data that support the findings of this study are available from the Key Laboratory of Adolescent Cyberpsychology and Behavior (CCNU), but restrictions apply to the availability of these data, which were used under license for the current study and so are not publicly available. Data are, however, available from the authors upon reasonable request and with permission of the Key Laboratory of Adolescent Cyberpsychology and Behavior (CCNU).

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
