# Peer review of "Nature Connectedness Reduces Internet Gaming Disorder: The Chain Mediating Role of Intolerance of Uncertainty and Desire Thinking"

_behavsci, 2024, doi:10.3390/bs14090844_

Round 1

Reviewer 1 Report

Comments and Suggestions for Authors

This paper offers a context-specific view on how nature connectedness has the potential to reduce internet gaming disorder among young people (n=571) in China. The study aimed to explore the relationship between nature connectedness and internet gaming disorder, and the mediating roles of intolerance of uncertainty and desire thinking. The study implemented a cross-sectional survey design and, informed by a detailed literature review, tested four hypotheses, namely:  

1) There is a negative correlation between nature connectedness and internet gaming disorder;

2) Nature connectedness negatively predicts internet gaming disorder via the mediating role of intolerance of uncertainty;

3) Nature connectedness negatively predicts internet gaming disorder via the mediating effect of desire thinking; and

4) Nature connectedness negatively predicts internet gaming disorder through the serial mediation chain from intolerance of uncertainty to desire thinking.  

Although the impact of green spaces (e.g., nature connectedness) has been reported before on mediating the negative consequences of internet addiction (including, internet gaming addiction), this study offers a context-specific view on the topic. This reviewer is not familiar with all internet gaming disorder-related research originating from China and can therefore not confirm the absolute originality of the study.  

The authors should be complimented on a paper with the logical flow of their arguments. The introduction contextualises the study and culminates in the problem statement, and offers the aim of the study. The literature review is detailed, informed by recent scholarly work, and as expected from a quantitative study, informes the four hypotheses outlined before. The literature review focuses on the following topics:  

1) Nature connectedness and internet gaming disorder This culminated in Hypothesis 1.  

2) Mediating role of intolerance of uncertainty. This culminated in Hypothesis 2.
  3) Mediating role of desire thinking This culminated in Hypothesis 3.
  4) Chain mediation from intolerance of uncertainty to desire thinking This culminated in Hypothesis 4.
  The materials and methods section is well structured. It is evident that a web-based survey was used to reach young people in the gaming environment to complete the questionnaire through non-probability sampling. Hence, the results cannot be generalised.The measures informing the study, namely the Inclusion of Nature in the Self Scale, Desire Thinking Questionnaire, Intolerance of Uncertainty Scale and Internet Gaming Disorder Scale-20 are adequately described for inclusion in the survey. In all instances, the validity and reliability is reported, where applicable. What is, however, not clear, is whether these measuring instruments are in the public domain, or whether the authors had to obtain permission to use the instruments.  

This reviewer is not an expert in statistics, but from the data analysis process, the tables and figures reported, the results could be understood and the conclusions reached seemed to flow logically. The discussion section of the manuscript is detailed, and it is indicated that hypothesis 1, 2 and 4 could be confirmed, while hypothesis 3 was rejected.  

The implications section starts to make recommendations to inform theory and practice. It should, however, be noted that some of these recommendations forwarded are known within the field of study and not necessarily original. The limitations of the cross-sectional survey are appropriately highlighted, and hence the authors make novel recommendations for future research.  

Some isolated problems with language are indicated via track changes. Overall, the technical care, tables, figures, and references used are acceptable.  

It is recommended that the manuscript is accepted for publication with minor corrections.

Comments on the Quality of English Language

Some suggestions with regard to the use of language:

- Use the acronym for internet gaming disorder (IGD) throughout the abstract and manuscript.

- Abstract: Should IGD be considered an "issue" or rather a mental disorder?

- Be careful to use the term 'addicts'; it is labeling and signals stigma.

- See the submission with comments for further consideration.

Reviewer 2 Report

Comments and Suggestions for Authors

1. The hypothesis derivation needs to be clearer. The author seems to be discussing the role of nature connectedness from the perspective of natural contact, but nature connectedness is not equal to natural contact. If the author wants to express that nature connectedness will increase natural contact and ultimately produce an effect, there needs to be a clearer narrative logic and more literature support. For example, this sentence needs to be supported by literature, "nature connectedness can promote individuals' contact with nature".

2. Regarding the limitations of the study, it is recommended to consider the measurement of nature connectedness. In addition to the single-question measurement method used in this study, there are other scales for nature connectedness. The author can also briefly describe the advantages and disadvantages of single-question measurement.

3. Can the author discuss the chain mediation effect based on the I-PACE model? It is recommended that the author consider whether Intolerance of Uncertainty and Desire Thinking can correspond to A and C in the I-PACE model.
